# HYBRID SPATIAL REPRESENTATIONS FOR SPECIES DISTRIBUTION MODELING

## ABSTRACT

We address an important problem in ecology called Species Distribution Modeling (SDM), whose goal is to predict whether a species exists at a certain position on Earth. In particular, we tackle a challenging version of this task, where we learn from presence-only data in a community-sourced dataset, model a large number of species simultaneously, and do not use any additional environmental information. Previous work has used neural implicit representations to construct models that achieve promising results. However, implicit representations often generate predictions of limited spatial precision. We attribute this limitation to their inherently global formulation and inability to effectively capture local feature variations. This issue is especially pronounced with presence-only data and a large number of species. To address this, we propose a hybrid embedding scheme that combines both implicit and explicit embeddings. Specifically, the explicit embedding is implemented with a multiresolution hashgrid, enabling our models to better capture local information. Experiments demonstrate that our results exceed other works by a large margin on various standard benchmarks, and that the hybrid representation is better than both purely implicit and explicit ones. Qualitative visualizations and comprehensive ablation studies reveal that our hybrid representation successfully addresses the two main challenges. Our code is open-sourced at `https://anonymous.4open.science/r/HSR-SDM-7360`.

## 1 INTRODUCTION

Understanding species distribution ranges is a key issue in ecological research, and it has become increasingly important in the context of the current global climate crisis and biodiversity decline. Conventionally, species distribution data has been collected through field studies by human experts and explorers, who must gather and assess large amounts of information to determine whether a species is present in a given region. These processes are typically slow and labor-intensive, and by the time the models are completed, they may already be outdated or irrelevant.

Species Distribution Modeling (SDM) is a method that aims to use collected data to directly predict the distribution range of species, thus making related ecological research easier (Elith & Leathwick, 2009; Elith et al., 2010; Miller, 2010). SDMs have many crucial applications in fields such as climate change assessment (Santini et al., 2021), invasive species management (Srivastava et al., 2019), and extinction risk mapping (Ramirez-Reyes et al., 2021). Whilst such models have achieved some success over the past two decades, most SDMs remain poor indicators of important ecological parameters (Lee-Yaw et al., 2022). Consequently, new SDM methodologies employing more advanced modeling techniques have continued to emerge (Beery et al., 2021).

One challenge in constructing SDMs is the collection of sufficient data for both training and testing (Feeley & Silman, 2011; Vaughan & Ormerod, 2005). The large volume of data required for constructing accurate models, coupled with the difficulty of obtaining it, has consistently been a major obstacle in the development of SDMs. With the emergence of community-sourced data platforms such as iNaturalist, eBird, and PlantNet, difficulties in data collection have been somewhat mitigated, but new challenges have arisen regarding data quality and the model's ability to process large volumes of data (Hartig et al., 2024). For example, due to the nature of the data collection process, most large species distribution datasets are highly susceptible to sampling bias, class imbalance, and noise (Benkendorf et al., 2023; Dubos et al., 2022; Kramer-Schadt et al., 2013).

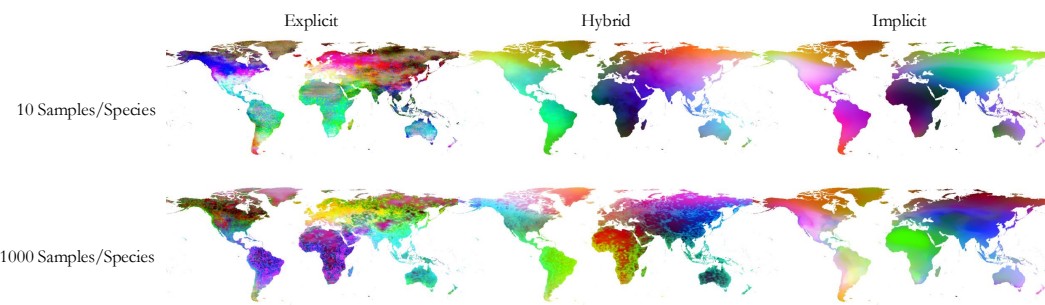

Figure 1: Results of Independent Component Analysis (ICA) on the feature embeddings for implicit, explicit, and hybrid models. The explicit embedding captures higher-frequency information and reflects local environmental data, while the implicit embedding, as a global location encoder, is less noisy. The hybrid representation combines the strengths of both. Differences are more pronounced in the 1000 samples per species setting. Note the noise in the explicit embeddings are mainly caused by the presence-only and community-sourced natures of our training data used.

Recent advances Cole et al. (2023) in using deep learning for SDMs has reduced the demand for large amounts of high-quality data. In particular, some recent methods applying implicit neural representations achieved considerable accuracy, and no longer required training signals besides presence-only data. However, in practice, predictions from those models are often of limited spatial precision due to the implicit nature of their representational schemes: neural networks inherently produce global embeddings that are not grounded in local features.

In this paper, we explore a challenging task that highlights the limitations of implicit representations. First, we use **presence-only data** instead of presence-absence data. Since confirming a species' presence is generally easier than confirming its absence, many previous studies have constructed SDMs using presence-only data (Barbet-Massin et al., 2012; Mac Aodha et al., 2019; Cole et al., 2023), making this a more difficult but valuable task. Second, we use **no additional environmental information**. Although conventional SDMs usually use a lot of environmental inputs, and satellite images are also a common source of information (Dollinger et al., 2024; Gillespie et al., 2024; He et al., 2015; Klemmer et al., 2023), we will focus on exploring the locational embedding and thus not use those information. Third, similar to most previous deep learning-based methods, we use iNaturalist, a **community-sourced dataset**, which, as previously discussed, presents various difficulties. Finally, we construct a single model for **a large number of species simultaneously**.

To address these issues, inspired by advances in explicit and hybrid representations, we propose a **Hybrid Spatial Representation** for Species Distribution Modeling. Specifically, our representation combines an implicit component based on FCNet (Mac Aodha et al., 2019) with an explicit component based on multiresolution hashgrids (Müller et al., 2022), forming a hybrid model well-suited for the SDM task. An intuitive view of these embeddings is shown in Figure 1.

Experiments show that our method achieves the best of both worlds, producing state-of-the-art results on this challenging task, outperforming both previously proposed methods and purely implicit or explicit versions of our model by significant margins. We also investigate the mechanisms behind the effectiveness of hybrid representations and characterize our model across a wide range of settings and evaluation methods for additional insights. Specifically, we show that hybrid representations are well-suited for learning from presence-only observations and modeling a large number of species simultaneously.

## 2 RELATED WORKS

### 2.1 SPECIES DISTRIBUTION MODELING

As discussed earlier, SDM is a challenging field that often requires learning from large volumes of inaccurate data. Recently, several works have used deep learning (Botella et al., 2018; Chen et al., 2017; Cole et al., 2023; Mac Aodha et al., 2019) to create SDMs from those massive datasets. This is

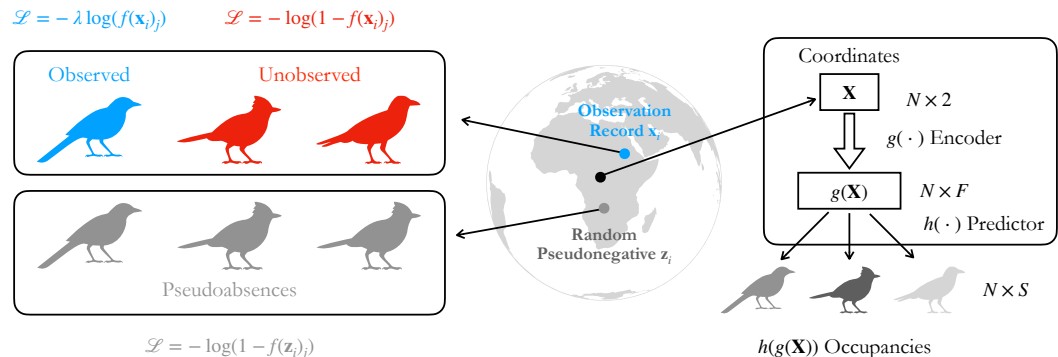

Figure 2: Illustration of our problem formulation and basic model structure.

a difficult task due to the inherent difficulties in the data, and therefore requires highly effective representational methods. The current state-of-the-art representation, as verified by several works (Cole et al., 2023; Lange et al., 2024; Rußwurm et al., 2023), is the FCNet architecture (Mac Aodha et al., 2019), which makes use of a Residual Network (He et al., 2016)-based structure to achieve effective implicit location embeddings.

It is understandable that explicit or hybrid representations have not been previously applied in SDMs, since — as our experiments will later demonstrate — explicit representations often produce noisy predictions with artifacts. However, there is a strong rationale for using explicit representations in SDMs: popular implicit embedding-based models, which rely on neural networks, struggle to capture local details. Since neural networks lack a separable component to describe local distributional features, predictions from implicit models often consist of blurry "blobs" of distributional peaks with poorly defined boundaries.

Our work demonstrates the power of combining implicit and explicit representations for SDM construction, and achieves state-of-the-art results on challenging benchmarks.

## 2.2 IMPLICIT, EXPLICIT, AND HYBRID REPRESENTATIONS

In fields such as signal processing, 3D vision, and computer graphics, Neural Implicit Representations (NIR) have achieved great results (Sitzmann et al., 2020; Mildenhall et al., 2021). A common pattern in those fields is that the success of implicit representations was often followed by the appearance of explicit ones which offer advantages such as increased accuracy and efficiency (Chen et al., 2022; Sun et al., 2022; Yu et al., 2021). In practice, hybrid representations that combine both implicit and explicit schemes often achieve superior performance by capturing the strengths of of both approaches.

While some previous works in the context of global spatial encoding have used hybrid or explicit representations (Kim et al., 2024; Mai et al., 2020; Rußwurm et al., 2023), to the best of our knowledge, no works on SDM have done this. In this work, inspired by Müller et al. (2022), we design an innovative two-dimensional multiresolution hashgrid as an explicit representation, specifically suited to the task of SDM construction. This model generates explicit embeddings, which are aggregated with conventional implicit embeddings from an FCNet-inspired portion. The aggregated embeddings are then put through a regular neural network as results. We demonstrate the merits of explicit representations, and show that our hybrid representation outperforms both implicit and explicit schemes by combining their advantages.

## 3 PRELIMINARIES

Let $\mathcal{P}(\cdot) : [-1, 1]^{n \times 2} \to \{0, 1\}^{n \times S}$ be the ground-truth presence function, where $S$ is the number of species in the model, and for the coordinates $(lat, lon)$ (regularized between -1 and 1), we have $\mathcal{P}([lat, lon])_i = 1$ iff the species $i$ is present at $(lat, lon)$. Let $\mathbf{X} \in [-1, 1]^{N \times 2}$ be a matrix of coordinates where observations have been performed, where $N$ is the number of observation entries.

Corresponding to the observations are species indices $\mathbf{s} \in [1, S]^N$, where $\mathcal{P}(\mathbf{x}_n)_i = 1$ if $i = s_n$. Here $\mathbf{x}_n$ represents the $n$th row of $\mathbf{X}$. While it is possible in practice that false presences occur through misidentification or aberrant migration, we treat this as regular noise in the data rather than a part of the problem formulation. Hence we need to construct a model $f(\cdot) : [-1, 1]^{n \times 2} \to \{0, 1\}^{n \times S}$ such that $f(\mathbf{X})$ approximates $\mathcal{P}(\mathbf{X})$.

This can be seen as a multiclass version of the positive-unlabeled learning problem, since observations can be seen as presences for one species and unlabeled data points for all others. Du Plessis et al. (2014) show that this is equivalent to a weighted positive-absence problem with the unlabeled points as absences, which coincides with works in SDM that consider "pseudoabsences" (Barbet-Massin et al., 2012). Works like Phillips et al. (2009) have used additional randomly sampled points as unlabeled points for all species.

Given random all-unlabeled pseudoabsences $\mathbf{Z} \in [-1, 1]^{N \times 2}$, the Assume Negative Full Loss as in Cole et al. (2023) can be written as follows:

$$\mathcal{L}_{\text{full}}(\mathbf{X}, \mathbf{s}, \mathbf{Z}) = -\frac{1}{NS} \sum_{i=1}^{N} \sum_{j=1}^{S} (\mathbb{1}_{j=s_i} \lambda \log f(\mathbf{x}_i)_j + \mathbb{1}_{j \neq s_i} \log(1 - f(\mathbf{x}_i)_j) + \log(1 - f(\mathbf{z}_i)_j)), \quad (1)$$

where subscripts represent row slices, and $\lambda$ is a hyper-parameter to prevent the latter two terms from dominating.

Under the NIR-based setting, the function $f(\mathbf{X})$ consists of two parts: a location embedding $g(\cdot) : [-1, 1]^{N \times 2} \to \mathbb{R}^{N \times F}$ (where $F$ is the dimension of the embedding, also known as the number of features), and the occupancy predictor $h(\cdot) : \mathbb{R}^{N \times F} \to \mathbb{R}^{N \times S}$ which takes the embeddings as input and outputs the occupancy predictions for the species. Hence the model is described by:

$$f(\mathbf{X}) = h(g(\mathbf{X})). \quad (2)$$

Usually $h(\cdot)$ has a simple structure, such as being a single linear layer, while $g(\mathbf{X})$ is a more sophisticated model. Hence the significance of the embedding $g$ lies in encoding the coordinates in such a manner that it would be easy for $h$ to conduct the final mapping step. We call the architecture of $g$ the *representation scheme*. The current state-of-the-art is FCNet (Mac Aodha et al., 2019), which makes use of a Linear-ReLU layer followed by four repetitions of Linear-ReLU-Dropout-Linear-ReLU residual blocks.

At the beginning of $g$ there is also often a quick positional encoding of the coordinates. This is because, per Sitzmann et al. (2020), implicit representations perform better when the inputs are of high frequency. FCNet uses the wrap encoding $(\sin(\pi lon), \cos(\pi lon), \sin(\pi lat), \cos(\pi lat))$.

Figure 2 displays a brief summary of the data and model structure in our problem formulation.

## 4 METHODS

### 4.1 MOTIVATION

We notice two insufficiencies in the implicit formulation above.

**Global Parameterization** Since $g$ is a neural network, in the back-propagation process, most parameters have nonzero gradient steps. More intuitively, one can see the implication that each parameter is equally capable of being associated with the Amazon Rainforest as with the Saharan Desert. In the process of training parameters gradually get implicitly mapped to different features, but there is still no guarantee that the parameters can reliably describe local environmental information.

**Low Signal Frequency** In addition, since MLPs follow the Lipschitz constraint, intuitively maintaining a degree of "smoothness," they often struggle to describe high-frequency patterns. Indeed, a lot of previous work in NIRs has focused on encoding the data to facilitate modeling. In our task, this means that embeddings for nearby locations tend to be similar regardless of their characteristics, which indicates an inability to describe local details. In practice this is very undesirable, since many ecological boundaries (*e.g.*, mountain ridges, wide rivers, and artificial constructs) cause sharp distinctions between ecosystems in physical proximity of each other.

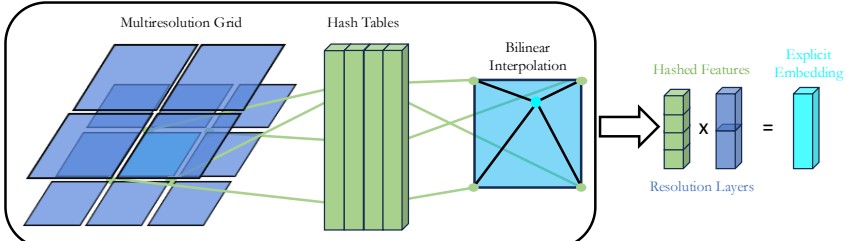

Figure 3: Illustration of our multiresolution hashgrid representation's mechanism. The Earth's surface is divided using several different resolutions into different levels, and vertices of the grids formed from division are mapped to hashed features in the hash table. The explicit embedding of any position (cyan) is calculated via calculating features (green) at each resolution level (blue) using bilinear interpolation and concatenating them.

**Principle for Explicit Embedding**  We propose a guiding principle which could simultaneously solve to problems above. We introduce explainable parameters which each correspond to only a specific region on Earth. If a data point is not contained within the region associated with a parameter, the gradient of that parameter with respect to this input is zero, thus creating *local* instead of *global* parameterization. In addition, at the boundaries between those regions, due to the change in the set of associated parameters, the Lipschitz constraint no longer applies, and thus the output can have arbitrarily high signal frequency.

Such an implementation would require an explicit rule dictating the correspondence between parameters and geographical regions, which currently no one has considered. Hence, we introduce a new explicit embedding scheme which suits our purposes.

### 4.2 MULTIRESOLUTION HASHGRIDS FOR SDMS

We propose using a multiresolution hashgrid encoding scheme as an explicit representation for SDM modeling. Specifically, we divide the Earth's surface into multiple grids with different resolution levels, and store trainable feature parameters associated with lattices of the grid in a hashtable. Embedded features on any given point for each resolution level are then calculated via bilinear interpolation. Finally, the output embedding is given by concatenating all hashed features from the different layers. An intuitive depiction is shown in Figure 3.

The resolution of each layer follows a geometric sequence from a maximum resolution to a minimum resolution (both of which are hyper-parameters). Specifically, given maximum and minimum resolutions $R_{max}$ and $R_{min}$, and a total of $L$ layers, the resolution of layer $l$ is calculated as follows:

$$R_l = R_{min} \exp(\frac{l}{L-1}(\log R_{max} - \log R_{min})) \qquad (3)$$

The grids allow for our model to explicitly capture information regarding the local environment by using the features as representation. Furthermore, the different resolutions allow for the description of explicit interactions in between smaller grids, based on their mutual intersection with larger grids. The hashgrid, meanwhile, ensures that the number of parameters created is not of overly large size.

### 4.3 HYBRID SPATIAL REPRESENTATION

Despite that our explicit embedding does carry many advantages, it inevitably also carries some drawbacks. For instance, due to the nature of encoding using artificially divided grids, there is a lot of noise in the resulting embedding. Furthermore, the implicit embedding's ability to aggregate global information is still valuable, while this ability is weakened for an explicit embedding. Hence, we propose a method to combine the two, and further, to make the model tunable between purely explicit and implicit encoding schemes.

Our method is the aggregate the locational embeddings from two parallel location encoders, one using the conventional implicit scheme and one using our explicit multiresolution hashgrid. We

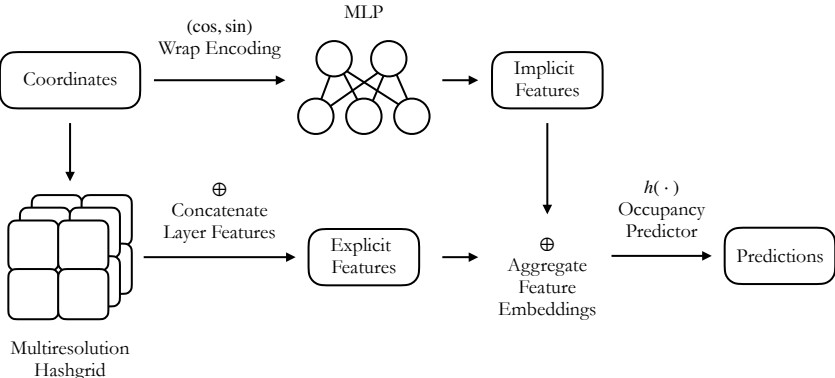

Figure 4: Simplified schematics of our hybrid SDM architecture. The implicit portion sends wrap-encoded coordinates into an MLP. The explicit portion uses the multiresolution hashgrid representation. The two feature embeddings are then concatenated and passed through the occupancy predictor.

concatenate resulting embeddings from the two, and proceed to input the concatenated results as the embedding for the occupancy predictor.

Letting $g_i(\cdot) : [-1, 1]^{N \times 2} \to \mathbb{R}^{N \times F_i}$ and $g_e(\cdot) : [-1, 1]^{N \times 2} \to \mathbb{R}^{N \times (F - F_i)}$ represent the encoders, the resulting hybrid model is:

$$f(\mathbf{X}) = h(g_i(\mathbf{X}) \oplus g_e(\mathbf{X})), \tag{4}$$

where $\oplus$ represents concatenation along the second dimension, and $F_i$ is the dimension of the implicit embedding. We refer to the ratio $\frac{F_i}{F}$ as "implicitness," and treat it as a tunable hyper-parameter. For an explicit model with $L$ layers and $M$ features per level, we have $F - F_i = L \times M$. In practice we keep $M$ to powers of two for implementational reasons and vary $L$ for tuning implicitness. A depiction of the schematics of our hybrid model is shown in Figure 4.

## 5 EXPERIMENTS

### 5.1 IMPLEMENTATION AND SETTINGS

Our codebase was altered from Cole et al. (2023), and we use their version of the FCNet encoding as well (which in turn is based on Mac Aodha et al. (2019)). We borrowed and altered part of the explicit encoding implementation from Tancik et al. (2023), which in turn used Müller (2021) as the core for the multiresolution hashgrid. All deep learning operations are based on PyTorch (Paszke et al., 2019), and all optimizers were Adam (Kingma & Ba, 2014).

All experiments were conducted on a single GPU: either an RTX A4000 (16GB) or a GeForce RTX 3090 (24GB). We run our models for 10 epochs each across learning rates 0.01, 0.003, 0.001, 0.0003, and 0.0001, and report the best results here. For most of the experiments below we also ran models across 9 implicitness settings from 0.0 to 1.0 with a step size of 0.125 but may have only reported some of the results here for clarity of presentation — for the full results, results under suboptimal learning rates, and raw data from graphs, please refer to Appendix A. For models with implicitness smaller than 0.5 we use $M = 16$, while for models with implicitness larger than 0.5 we use $M = 8$.

The dataset used was iNaturalist, which is a popular choice for benchmarking SDMs and has been used in multiple previous works (Mac Aodha et al., 2019; Cole et al., 2023; Rußwurm et al., 2023). We use standardized settings introduced by Cole et al. (2023) in order to ensure fair comparison, and the specific version of the iNaturalist dataset we used was the same as theirs as well. To deal with class imbalance, as well as to condition on the amount of data provided, we sample 10, 100, or 1000 observations per species from the dataset, referred to below as the "Observation Cap" or "Obs. Cap" for short. The data, and thus the model, covers a total of 47375 species.

It should be noted that the purely implicit version of our model, which uses FCNet (Mac Aodha et al., 2019), is architecturally equivalent to SINR (Cole et al., 2023), and differences between results from the two are attributable to hyper-parameter tuning and randomness.

Table 1: Results of experiments on S&T and IUCN benchmarks. Reported values are mAP percentages. Values in parentheses represent implicitness. Values for SINR, GP, and BDS are as reported by Cole et al. (2023). Results for BDS used all training data with no observation cap. We achieve large improvements compared to previous works, especially on the difficult IUCN task.

| Benchmark | S&T | | | IUCN | | |
|---|---|---|---|---|---|---|
| Obs. Cap | 10 | 100 | 1000 | 10 | 100 | 1000 |
| Ours-Explicit (0.0) | 60.21 | 71.23 | 76.01 | 48.83 | 62.46 | 64.23 |
| Ours-Hybrid (0.25) | **66.76** | 75.05 | 77.86 | 58.30 | 69.02 | 68.03 |
| Ours-Hybrid (0.5) | 66.64 | **75.27** | **78.47** | **59.39** | **69.57** | **70.32** |
| Ours-Hybrid (0.75) | 66.54 | 75.01 | 78.01 | 58.28 | 69.23 | 69.46 |
| Implicit (1.0) | 65.59 | 73.12 | 76.81 | 50.98 | 62.06 | 65.57 |
| SINR (Cole et al., 2023) | 65.36 | 72.82 | 77.15 | 49.02 | 62.00 | 65.84 |
| GP (Mac Aodha et al., 2019) | | | 73.14 | | | 59.51 |
| BDS (Berg et al., 2014) | | | 61.56* | | | 37.13* |

Table 2: Results of experiments on the GeoFeature benchmark. Reported values are averaged $R^2$ correlations across eight environmental features. Values for SINR and GP are as reported by Cole et al. (2023). As shown, explicit models are the most correlated with environmental features.

| | Ours (Implicitness) | | | | | | |
|---|---|---|---|---|---|---|---|
| Obs. Cap | 0.0 | 0.25 | 0.5 | 0.75 | 1.0 | SINR | GP |
| 10 | **74.6** | 74.5 | 73.5 | 72.5 | 71.1 | 71.2 | |
| 100 | **78.0** | **78.0** | 77.1 | 76.6 | 73.9 | 73.6 | |
| 1000 | **79.3** | 79.0 | 78.6 | 77.9 | 75.2 | 75.2 | 72.4 |

## 5.2 BENCHMARKING

**Comparison on SDM Benchmarks**   We evaluate our models on two human expert-created distribution range datasets: S&T (eBird Status and Trends) and IUCN (the International Union for Conservation of Nature). There are a total of respectively 535 and 2418 species overlapping between the iNaturalist training dataset and the two testing datasets, and we report the Mean Average Precision (mAP) of the models' predictions. Results are shown in Table 1. The reported results for all our models are means from five repetitions: for error bars please refer to Subsection 5.3, and for the full raw data please refer to Appendix A.

As shown, our models consistently achieve state-of-the-art results on those standardized tasks, by margins of up to **21.2%** relative improvement (for few-shot learning with 10 samples/species on the difficult IUCN benchmark), demonstrating the benefit of using a hybrid representation. In addition, we see that the model with implicitness 0.5 performs well across all scenarios, ruling out the need for extensive tuning of the implicitness hyper-parameter in practice (discussed more in Subsection 5.3).

**Correlation with Environmental Data**   We investigate whether our explicit representation indeed represents local environmental information better as expected by using some common environmental parameters as proxies. The data comes from the GeoFeature benchmark in Cole et al. (2023), and includes 8 parameters in different locations sampled within the contiguous United States, such as above-ground carbon, elevation, *etc*. We report the average $R^2$ correlation between the embeddings and the environmental parameter. Results are shown in Table 2.

As shown, explicit models are the most correlated with environmental information, as we expected in our design. There is also a very clear negative relation between implicitness and the performance on this task. We conclude that explicit models have strong capability for capturing environmental information. Interestingly, this might indicate that in hybrid models, the explicit portion of the embedding is serving as a "bootstrapped" proxy for environmental data, thus improving the results.

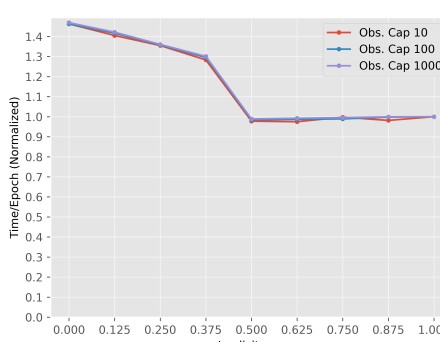 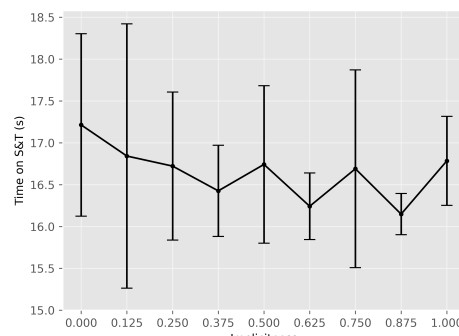

Figure 5: The training time per epoch of different models, divided by the training time of the implicit model for normalization.

Figure 6: The inference time of models on the S&T baseline. Error bars are ± one standard deviation across five repetitions.

Table 3: All were run with an observation cap of 1000. The hybrid model had implicitness 0.5. Results display that simply increasing the dimension of the implicit embedding (and thus the number of parameters) does not result in results comparable to our hybrid method.

|                    | Number of Features (Implicit) | | | | |
|                    | 256 (SINR) | 512 | 1024 | 2048 | Ours (Hybrid) |
| --- | --- | --- | --- | --- | --- |
| Training Time (s)  | 574  | 839   | 1376  | 2760          | **568**  |
| Inference Time (s) | 16.8 | 23.2  | 24.7  | 27.7          | **16.7** |
| S&T mAP (%)        | 77.15       | 77.73 | 78.02 | 78.29  | **78.47** |

**Training and Inference Speed**  We trained models for all 9 implicitness settings for a single epoch on single RTX A4000 (16GB) GPUs, running five repetitions simultaneously. We then conduct inference under the same settings on the S&T benchmark. We found that all standard deviations for the training time are within 2% times the mean, suggesting that the results have high statistical significance, so we omit reporting them and just report the means here. Results are shown in Figures 5 and 6.

As shown, we see that when implicitness is less than 0.5 (the model leans explicit), there seems to be a training overhead of up to around 47% times the implicit model, presumably due to the larger number of features per level. However, for models with implicitless greater than or equal to 0.5 (the model leans implicit), there is no overhead compared to the implicit model. Hence, using our hybrid model for better results does not require incurring sacrifices in speed. Meanwhile, no statistically significant difference between the models was observed for inference time.

**Comparison with Larger Implicit Networks**  In all of the experiments above, the dimension of the location embedding is 256 as in SINR. One may wonder whether simply increasing the dimension of this feature embedding would allow the resulting "fat" implicit model to achieve better results than our explicit representation. We conduct experiments to show that this is not the case: while marginal performance gains can be achieved, they come at a very heavy cost for training and inference speed, and still cannot exceed results of our hybrid model. Results are reported in Table 3.

## 5.3 CHARACTERIZATION

**Hyper-Parameter Sensitivity Analysis of Implicitness**  Here we present data from all 9 implicitness settings under the two benchmarks, each ran for five repetitions to rule out the effect of randomness. All reported results are under the best learning rate settings for respective models. Results are displayed in Figure 7.

Our results further verify that any value of implicitness except 0.0 and 1.0 (the degenerate cases) are relatively insensitive and robust: all hybrid models have several standard deviations' improvement

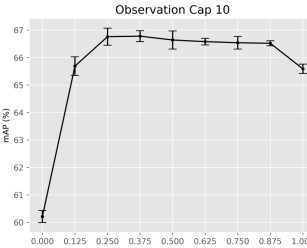 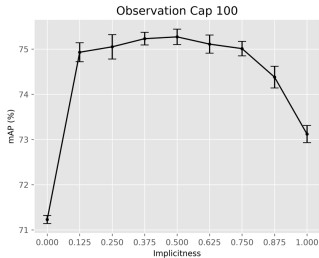 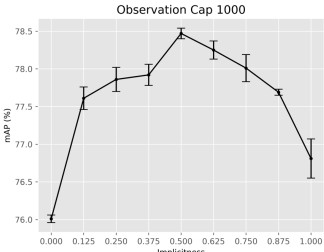

Figure 7: Results of models with different implicitness on the S&T task. Error bars are ± one standard deviation across five repetitions. As shown, implicitness is not a very sensitive hyper-parameter.

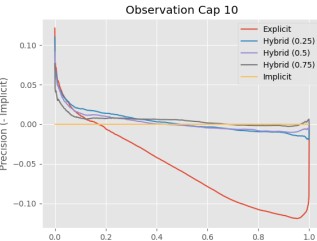 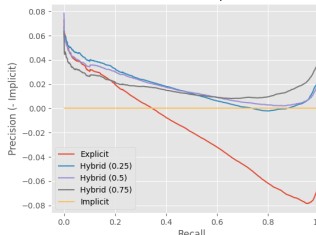 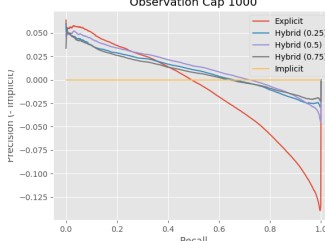

Figure 8: Precision-Recall Curves averaged across the 535 species on the S&T benchmark. Precision is regularized with respect to the implicit models. As shown, models with lower implicitness have higher precision in low-recall scenarios, and vice versa.

compared to explicit or implicit ones. Hence, no extensive tuning is needed for this newly introduced hyper-parameter. In practice, we recommend a simple value of 0.5.

**Precision-Recall Trade-Off**    To further identify mechanisms via which the hybrid model achieves superior results, we plot the Precision-Recall Curves (PRCs) for models with 0.0, 0.25, 0.5, 0.75, and 1.0 implicitness here. We use PRCs instead of other tools like ROCs because it is better suited to our task, which has strong class imbalance (Saito & Rehmsmeier, 2015). Results are shown in Figure 8.

It can be seen that the improvement our models achieve stem mainly from their high precision for low-recall scenarios in comparison to implicit models. However, the precision of explicit models plunge when recall is high. Hybrid models successfully balance between the two, achieving the best of both worlds.

**Conditioning on the Number of Species**    To verify that hybrid and explicit models are better at aggregating information from the distribution of multiple species, we run the S&T and environmental data baselines again with different numbers of species. Following Cole et al. (2023)'s approach, we train the models on the 535 S&T species only first, and increment the number of species in intervals of 4000. All models in this experiment were trained with an observation cap of 1000 observations per species. Results are shown in Figures 9 and 10.

We can notice from the results that on the S&T benchmark, hybrid models perform better than others, and that this gap generally tends to grow as the number of species increases, suggesting the superiority of hybrid models in terms of learning distributions across species. The implicit baseline, meanwhile, learns only a limited amount of new information from having more species modeled. However, explicit models perform poorly, especially with large amounts of species — is this due to their own incapability or to overfitting? Our experiments on correlation with environmental data show that it is the latter, as explicit models are actually much better at inferring environmental information from data on large numbers of species, and present larger performance gains in this respect when the number of species increases.

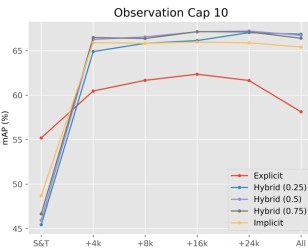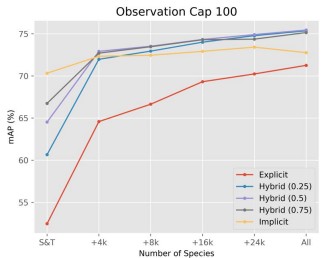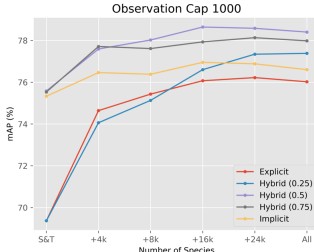

Figure 9: Results on the S&T benchmark with respect to the number of species. Hybrid models experience larger increases in performance as the number of species increase.

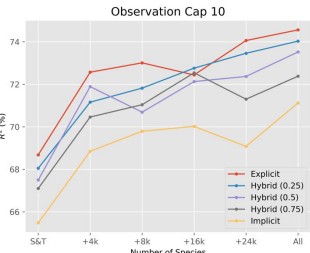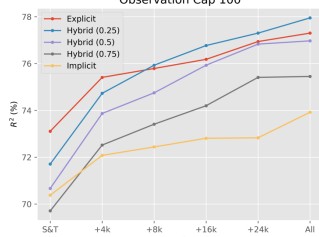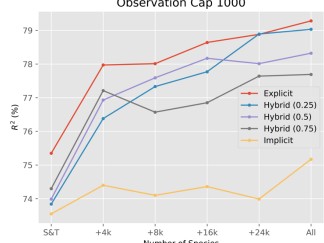

Figure 10: Results on the GeoFeatures benchmark with respect to the number of species. Explicit models experience larger increases in performance as the number of species increase, due to their inherent ability of inferencing environmental information from species observations.

## 6 CONCLUSION

Our work explores the application of explicit and hybrid representations to the task of SDM construction. We introduce an innovative explicit representation scheme, and use it in conjunction with conventional implicit methods to form a hybrid representation. Experiments show that our hybrid models consistently achieve state-of-the-art accuracy on multiple standard benchmarks, outperforming both implicit and explicit models, and that our explicit representations are good at representing local environmental information. We also conducted extensive experiments to characterize our models and investigate their properties.

As supported by our experiments, we conclude that the hybrid representation is well- suited to our problem formulation because:

- **Presence-Only Data**: The hybrid model presents informative embeddings that have both low noise and high signal frequency, thus allowing accurate and precise inference based on presence-only data.

- **No Additional Environmental Information**: The explicit portion of the model can bootstrap environmental information from the species presences, thus providing a proxy for environmental data.

- **Community-Sourced Dataset**: Keeping the implicit portion allows for reduction of noise and bias from the dataset, preventing the explicit portion from overfitting to inaccuracies.

- **Large Number of Species**: As the number of species increases, the hybrid representation consistently gains performance, while the explicit representation (although prone to overfitting) becomes better at inferring environmental information.

**Limitations**   Since our work concentrates on effective representations for the locational embedding portion of SDMs, we did not incorporate extra components such as remote sensing data, presence-absence data, or environmental data. We consider those extensions to our model interesting directions for future research.

**Ethics Statement**   Our work, like most similar ones on SDMs, is prone to the ethical hazards of damaging conservation fairness (Donaldson et al., 2016; Fedriani et al., 2017), insufficient reliability (Lee-Yaw et al., 2022), and potential for unintended uses such as poaching (Atlas & Dando, 2006). We suggest judicious use of our methods and careful interpretation of results.

**Reproducibility Statement**   We open-sourced our implementation as a codebase, allowing all of our experimental results to be easily reproducible and making it easy for others to extend upon our work. We also released a zoo of all trained models under different learning rates, implicitness, and observation caps, such that the experiments can be repeated with minimal difficulty.

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

## A    FULL RESULTS AND RAW DATA

**Benchmarking with Different Implicitness Settings**    Shown in Tables 4-6.

**Training and Inference Time**    Shown in Tables 7 and 8.

**Searching for Optimal Learning Rate**    Shown in Tables 9-11.

Table 4: Full results on S&T benchmark.

| Implicitness | Obs. Cap | | |
|---|---|---|---|
| | 10 | 100 | 1000 |
| 0.0 | 60.21±0.22 | 71.23±0.09 | 76.01±0.05 |
| 0.125 | 65.69±0.34 | 74.93±0.21 | 77.61±0.15 |
| 0.25 | 66.76±0.31 | 75.05±0.27 | 77.86±0.16 |
| 0.375 | **66.78**±0.20 | 75.23±0.14 | 77.92±0.14 |
| 0.5 | 66.64±0.33 | **75.27**±0.17 | **78.47**±0.07 |
| 0.625 | 66.58±0.12 | 75.11±0.20 | 78.25±0.12 |
| 0.75 | 66.54±0.23 | 75.01±0.16 | 78.01±0.18 |
| 0.875 | 66.52±0.09 | 74.38±0.24 | 77.69±0.04 |
| 1.0 | 65.59±0.17 | 73.12±0.19 | 76.81±0.26 |

Table 5: Full results on IUCN benchmark.

| Obs. Cap | Implicitness | | | | | | | | |
|---|---|---|---|---|---|---|---|---|---|
| | 0.0 | 0.125 | 0.25 | 0.375 | 0.5 | 0.625 | 0.75 | 0.875 | 1.0 |
| 10 | 48.83 | 56.39 | 58.30 | **59.56** | 59.39 | 58.29 | 58.28 | 57.52 | 50.98 |
| 100 | 62.42 | 66.71 | 69.02 | 69.41 | **69.57** | 68.95 | 69.23 | 67.69 | 62.06 |
| 1000 | 64.23 | 67.57 | 68.36 | 69.13 | **70.32** | 70.27 | 69.46 | 68.27 | 65.57 |

Table 6: Full results on GeoFeatures benchmark.

| Obs. Cap | Implicitness | | | | | | | | |
|---|---|---|---|---|---|---|---|---|---|
| | 0.0 | 0.125 | 0.25 | 0.375 | 0.5 | 0.625 | 0.75 | 0.875 | 1.0 |
| 10 | 74.56 | **74.65** | 74.51 | 74.22 | 73.52 | 72.98 | 72.54 | 72.58 | 71.12 |
| 100 | 78.04 | **78.11** | 77.95 | 77.80 | 77.13 | 76.67 | 76.62 | 76.42 | 73.92 |
| 1000 | **79.28** | 79.19 | 79.03 | 78.87 | 78.59 | 78.12 | 77.88 | 77.48 | 75.17 |

Table 7: Full results on training time.

| Implicitness | Obs. Cap | | |
|---|---|---|---|
| | 10 | 100 | 1000 |
| 0.0 | 26.77±0.04 | 237.99±0.27 | 843.79±1.31 |
| 0.125 | 25.71±0.07 | 230.46±0.62 | 816.49±1.49 |
| 0.25 | 24.79±0.19 | 220.90±0.59 | 781.41±1.77 |
| 0.375 | 23.48±0.08 | 210.75±0.72 | 747.15±2.40 |
| 0.5 | 17.89±0.15 | 160.10±0.45 | 567.90±1.82 |
| 0.625 | 17.84±0.04 | 160.52±0.46 | 570.05±2.04 |
| 0.75 | 18.26±0.34 | 160.93±0.55 | 571.61±1.73 |
| 0.875 | 17.96±0.05 | 162.32±0.64 | 573.93±1.10 |
| 1.0 | 18.30±0.16 | 162.71±1.10 | 574.37±2.34 |

Table 8: Full results on inference time.

| Implicitness | Time (s) |
|---|---|
| 0.0 | 17.21±1.09 |
| 0.125 | 16.84±1.58 |
| 0.25 | 16.72±0.88 |
| 0.375 | 16.43±0.54 |
| 0.5 | 16.74±0.94 |
| 0.625 | 16.24±0.40 |
| 0.75 | 16.69±1.18 |
| 0.875 | 16.15±0.25 |
| 1.0 | 16.78±0.53 |

Table 9: Full results on S&T benchmark with different learning rates.

| Obs. Cap | Learning Rate | Implicitness | | | | | | | | |
|---|---|---|---|---|---|---|---|---|---|---|
| | | 0.0 | 0.125 | 0.25 | 0.375 | 0.5 | 0.625 | 0.75 | 0.875 | 1.0 |
| 10 | 0.01 | 42.29 | 40.95 | 42.66 | 43.49 | 44.87 | 46.87 | 22.78 | 54.48 | 64.52 |
| | 0.003 | **60.07** | 58.05 | 59.09 | 60.30 | 59.98 | 60.51 | 62.51 | 63.80 | 64.92 |
| | 0.001 | 58.13 | **64.57** | **66.86** | **67.00** | **66.73** | **66.46** | **66.40** | **66.58** | **65.41** |
| | 0.0003 | 24.98 | 57.13 | 63.21 | 64.18 | 64.97 | 65.47 | 65.30 | 65.15 | 64.46 |
| | 0.0001 | 24.42 | 34.12 | 44.49 | 47.93 | 50.78 | 52.98 | 53.43 | 55.38 | 57.91 |
| 100 | 0.01 | 48.21 | 50.40 | 50.97 | 52.32 | 57.32 | 59.12 | 23.50 | 29.34 | 70.73 |
| | 0.003 | 58.77 | 60.87 | 61.55 | 63.26 | 65.73 | 67.78 | 69.00 | 71.39 | 72.53 |
| | 0.001 | 70.27 | 70.64 | 71.62 | 72.52 | 73.23 | 73.44 | 73.72 | 74.13 | **73.17** |
| | 0.0003 | **71.25** | **74.98** | **75.35** | **75.53** | **75.44** | **75.24** | **75.14** | **74.30** | 72.76 |
| | 0.0001 | 64.05 | 71.08 | 72.13 | 72.12 | 72.25 | 72.43 | 72.46 | 71.95 | 71.26 |
| 1000 | 0.01 | 60.21 | 62.17 | 63.54 | 64.71 | 68.24 | 69.33 | 19.28 | 19.28 | 71.75 |
| | 0.003 | 65.25 | 66.99 | 68.07 | 68.58 | 72.12 | 72.26 | 73.81 | 74.44 | 75.78 |
| | 0.001 | 72.20 | 73.52 | 74.07 | 74.29 | 76.16 | 76.03 | 75.98 | 76.63 | **77.10** |
| | 0.0003 | **76.02** | **77.44** | 77.38 | 77.88 | **78.40** | **78.15** | **77.98** | **77.81** | 76.60 |
| | 0.0001 | 74.83 | 60.21 | **78.08** | **77.91** | 78.13 | 77.91 | 77.56 | 76.89 | 75.41 |

Table 10: Full results on IUCN benchmark with different learning rates.

| Obs. Cap | Learning Rate | Implicitness | | | | | | | | |
|---|---|---|---|---|---|---|---|---|---|---|
| | | 0.0 | 0.125 | 0.25 | 0.375 | 0.5 | 0.625 | 0.75 | 0.875 | 1.0 |
| 10 | 0.01 | 33.60 | 32.71 | 34.31 | 35.70 | 36.00 | 37.63 | 0.86 | 44.39 | 47.71 |
| | 0.003 | **48.83** | 52.22 | 53.32 | 54.41 | 53.79 | 54.42 | 55.25 | 56.16 | 49.87 |
| | 0.001 | 38.41 | **56.39** | **58.30** | **59.56** | **59.39** | **58.29** | **58.28** | **57.52** | **50.98** |
| | 0.0003 | 4.82 | 34.12 | 46.36 | 50.72 | 53.53 | 54.02 | 54.28 | 52.99 | 46.85 |
| | 0.0001 | 1.18 | 8.02 | 15.12 | 19.53 | 24.58 | 31.42 | 31.24 | 31.82 | 31.85 |
| 100 | 0.01 | 32.51 | 34.96 | 36.98 | 38.44 | 43.77 | 45.82 | 0.88 | 0.85 | 55.89 |
| | 0.003 | 48.67 | 49.54 | 51.09 | 53.05 | 56.54 | 58.19 | 60.54 | 61.77 | 58.52 |
| | 0.001 | **62.42** | 64.27 | 64.79 | 65.49 | 67.33 | 67.38 | 67.12 | 66.44 | 60.54 |
| | 0.0003 | 61.91 | **66.71** | **69.02** | **69.41** | **69.57** | **68.95** | **69.23** | **67.69** | **62.06** |
| | 0.0001 | 46.58 | 59.67 | 63.88 | 64.98 | 65.41 | 65.26 | 64.87 | 63.39 | 58.77 |
| 1000 | 0.01 | 35.99 | 38.95 | 40.47 | 42.25 | 48.24 | 50.42 | 1.00 | 0.85 | 54.01 |
| | 0.003 | 44.36 | 46.05 | 48.04 | 49.62 | 54.64 | 56.57 | 58.19 | 60.05 | 59.31 |
| | 0.001 | 58.68 | 59.70 | 61.11 | 61.37 | 64.45 | 64.90 | 66.08 | 65.43 | 64.40 |
| | 0.0003 | **64.23** | **67.57** | 68.03 | 68.56 | **70.32** | **70.27** | **69.46** | **68.27** | **65.57** |
| | 0.0001 | 60.73 | 66.53 | **68.36** | **69.13** | 69.50 | 68.41 | 68.83 | 67.15 | 62.30 |

Table 11: Full results on GeoFeatures benchmark with different learning rates.

| Obs. Cap | Learning Rate | Implicitness | | | | | | | | |
|---|---|---|---|---|---|---|---|---|---|---|
| | | 0.0 | 0.125 | 0.25 | 0.375 | 0.5 | 0.625 | 0.75 | 0.875 | 1.0 |
| | 0.01 | 73.54 | 73.64 | 72.99 | 73.14 | 72.65 | 72.47 | 55.48 | 70.19 | 69.78 |
| | 0.003 | **74.56** | **74.65** | 74.03 | **74.22** | **73.52** | **72.98** | 72.38 | **72.58** | **71.12** |
| 10 | 0.001 | 73.73 | 74.43 | **74.51** | 73.50 | 73.28 | 72.69 | **72.54** | 71.34 | 70.76 |
| | 0.0003 | 70.10 | 71.86 | 72.64 | 72.57 | 72.68 | 72.09 | 71.85 | 70.73 | 69.83 |
| | 0.0001 | 69.98 | 70.65 | 70.64 | 70.26 | 70.80 | 70.64 | 68.94 | 69.89 | 67.59 |
| | 0.01 | 73.22 | 73.37 | 72.79 | 72.08 | 72.17 | 72.46 | 59.01 | 53.63 | 67.81 |
| | 0.003 | 75.90 | 76.61 | 75.54 | 75.65 | 76.00 | 74.54 | 75.12 | 74.28 | 71.01 |
| 100 | 0.001 | **78.04** | 78.05 | 77.25 | 77.59 | **77.13** | 76.64 | **76.62** | 76.42 | 73.66 |
| | 0.0003 | 77.30 | **78.11** | **77.95** | **77.80** | 76.97 | **76.67** | 75.45 | 76.12 | **73.92** |
| | 0.0001 | 74.53 | 75.75 | 76.66 | 76.24 | 75.68 | 75.26 | 73.89 | 73.72 | 71.93 |
| | 0.01 | 74.10 | 74.98 | 74.50 | 74.93 | 74.32 | 74.45 | 62.69 | 53.68 | 62.99 |
| | 0.003 | 76.31 | 76.84 | 76.87 | 76.79 | 76.29 | 76.28 | 76.32 | 75.14 | 72.33 |
| 1000 | 0.001 | 78.98 | 78.75 | 78.74 | 78.52 | **78.59** | 77.36 | **77.88** | 77.44 | 74.82 |
| | 0.0003 | **79.28** | **79.19** | **79.03** | **78.87** | 78.32 | **78.12** | 77.69 | **77.48** | **75.17** |
| | 0.0001 | 78.80 | 78.45 | 78.91 | 78.58 | 78.42 | 77.89 | 77.17 | 76.80 | 74.18 |