# OpenReview forum: "Hybrid Spatial Representations for Species Distribution Modeling"
_ICLR.cc/2025/Conference — ICLR 2025 Conference Withdrawn Submission_

### Official Review · Reviewer_YuGu · 2024-10-28

**Soundness:** 4
**Presentation:** 4
**Contribution:** 3
**Rating:** 8
**Confidence:** 4

**Summary:**

This paper tackles Species Distribution Modeling (SDM), aiming to predict species' geographic ranges. Building upon the success of Neural Implicit Representations (NIR) in SDM, the authors propose a novel hybrid model that integrates an explicit representation of location with the existing implicit approach. Specifically, they introduce a multiresolution hashgrid where grid lattices are mapped to learned embeddings in a hashtable. This explicit component complements the global nature of NIR, allowing the model to capture local environmental variations. Experiments, conducted on the same benchmark used to evaluate NIR, demonstrate that this hybrid model achieves state-of-the-art performance, outperforming both purely implicit and explicit models, in the challenging setting where no additional environmental variables are provided. The results highlight the benefit of combining implicit and explicit representations for enhanced spatial precision in SDM.

**Strengths:**

Originality: The paper demonstrates originality by introducing a hybrid implicit-explicit representation to the area of Species Distribution Modeling (SDM), an underexplored area in machine learning and where the proposed method, using a multiresolution hashgrid for the explicit component, is new.

Quality: The paper presents a well-executed experimental evaluation. The core claim of improved performance over the prior NIR model is supported by results on established benchmarks (S&T and IUCN). Importantly, the experiments varying the "implicitness" parameter provide compelling evidence for the advantages of the hybrid approach. The inclusion of error bars in the appendix for the S&T benchmark further strengthens the validity and reliability of the reported improvements.

Clarity: The paper is generally well-written and clearly explains the proposed method. The illustrations (Figures 1, 2, 3, and 4) are helpful in visualizing the concepts and the overall model architecture. The problem formulation and the motivation for the hybrid approach are also well-articulated. The paper clearly situates itself within the existing literature on both SDM and implicit/explicit representations (I discovered some related references I personally wasn’t aware of, in fact).

Significance: The paper addresses a problem of practical ecological significance, namely, efficiently and accurately modeling species distributions. The demonstrated improvements over the state-of-the-art in the absence of environmental variables makes for a significant contribution worth highlighting at ICLR. The fact that the hybrid model is computationally comparable to the previous SINR model increases its potential for adoption.

**Weaknesses:**

A clear limitation of the work is the decision to not rely on environmental variables. Generally, it doesn’t seem like there is a good reason not to use such additional inputs in practice. Notably, MaxEnt models leverage such information and are often very hard to beat in this application domain (see [1] where it in fact appears to be SOTA). While I think studying the setting without environmental variables is interesting and can be sufficient for a publication, this does reduce the potential for impact of this work significantly.

While computationally the proposed method is doing well, there is also no discussion on its memory footprint, which I suspect to be substantially higher than SINR. While I suspect it is still reasonable and manageable, it would be preferable for this aspect of the method to be discussed explicitly. Indeed, in a way what the proposed method is doing is that it is trading against the compute cost of using a fully explicit model with much larger hidden representation, with Table 3 suggesting that a fully implicit approach would be able to reach the performance of the proposed hybrid approach by using more features. Discussing how to think about this tradeoff would have made the submission better.

Finally, here are a couple minor typos:
  * "Our method is the aggregate the locational embeddings" => Our method is *to* aggregate the locational embeddings
  * "but may have only reported" => but have only reported

[1] On the selection and effectiveness of pseudo-absences for species distribution modeling with deep learning (https://arxiv.org/abs/2401.02989)

**Questions:**

For some reason, there are no error bars results for IUCN and GeoFeatures (that I could find). Am I missing something or are they indeed not provided? A clarification would be appreciated.

I also could not find details on how hyper-parameters were tuned, notably on the details of the holdout set that was used. Could the authors better detail how model selection was performed?

---

### Official Review · Reviewer_4HPc · 2024-10-29

**Soundness:** 2
**Presentation:** 2
**Contribution:** 2
**Rating:** 3
**Confidence:** 4

**Summary:**

This paper focuses on the task of species distribution modelling using citizen scientist collected “presence only” data. Only location data is used for this task rather than local environmental features as is often used in the literature.

The authors introduce a new approach that combines a neural network-based “implicit” state of the art model for this task (i.e. SINR from Cole et al ICML 2023) with an “explicit” method that uses a multi-resolution hash grid, taking inspiration from a recent approach used to create neural graphics primitives.

Location features produced using both of these methods are concatenated to produce a “hybrid” representation which is demonstrated  to be more effective for species range estimation than the features produced using either method separately.

The authors show that this hybrid approach outperforms the existing SOTA SINR approach on a set of benchmark evaluation tasks. They perform further experiments to show the role of the “implicit” and “explicit” parts of their approach.

**Strengths:**

The main strengths are:

[S1] The authors introduce an effective “hybrid” approach for machine learning-based species range estimation. This approach combines the current state of the art “implicit” approach for SDM with a recent “explicit” method for producing neural graphics primitives. This method beats the current state of the art on a set of benchmark evaluation tasks for species range estimation.

[S2] Extensive experiments are performed to investigate the impact on performance of several factors such as the training data used, the number of species modelled, and the “implicitness” of the model. Additional interesting experiments focusing on the “implicitness” of the model and the impact of the size of the location embedding in SINR models are also included.

[S3] The related work clearly describes most of the recent advances in machine learning approaches to species distribution modelling. The preliminaries section is clear and understandable.

**Weaknesses:**

The main weaknesses are:

[W1] Significance of contribution. L239 indicates that correspondence between learned parameters and geographic space is a novel idea. However many geospatial models exist that make use of an “explicit” or “hybrid” approach. One example is Kim et al’s “Hybrid Neural Representations for Spherical Data” from ICML 2024, which is cited but only briefly mentioned on L148. Kim et al also make use of parameters aligned with multiple grids at different resolutions over the globe. While no hash table is used, the “explicit” approach is conceptually very similar. Müller et al’s “Instant Neural Graphics Primitives with a Multiresolution Hash Encoding” from Transactions on Graphics 2022 which the authors cite as the main inspiration for their hash encoding, though these are not focussed on geospatial modelling. In Müller et al’s NeRF experiments (see Section 5.4 Model Architecture) they describe a hybrid-like approach that uses hash extracted features for the density and a more implicit style MLP for the color.
On L149, in the context of  hybrid or explicit representations for species distribution modelling, the text states that “to the best of our knowledge, no works on SDM have done this”. However, Rußwurm et al., ICLR 2024 actually perform experiments in Table 1 (c) of their paper where they estimate the range of different species and measure the effectiveness of the predicted ranges in the task of assisting image classification. However, they do not quantitatively evaluate the quality of the predicted ranges directly.

[W2] Potentially overfitting to the test set. Performance on the evaluation tasks is used to select a hyperparameter (the learning rate) for the results in Tables 1 and 2. From checking the result in Tables 9, 10 and 11 in the appendix, it appears that even within a given “implicitness” setting, different hyperparameters have been selected for different evaluation tasks, assumedly requiring new models to be trained specifically for each evaluation task. This appears to be a significant issue and could lead to inflated results from overfitting your approach to the evaluation tasks.

[W3] Unclear methodology. The paper lacks sufficient details on the “explicit” approach, making it challenging to understand the method's implementation. Specifically, it is unclear how the different grids are “stacked.” It is not specified whether grids at different resolutions share lattice points, and if so, whether they share the same feature representation for those points at different resolutions. There is also no explanation of whether an offset is applied to the different resolution grids to prevent them from aligning and potentially causing artifacts in the resultant range map. Additionally, Figure 3, which is intended to explain the explicit method, is difficult to interpret. On the right-hand side of the figure, it is unclear why the green “Hashed Features” cuboid is being multiplied by the blue “Resolution Layers” cuboid. It is also unclear why the “Hashed Features” shape is composed of 4 cubes while the “Resolution Layers” shape is only composed of 2. As a minor point, the cyan square in the figure is very difficult to spot because it is obscured by semi-transparent blue squares, which also makes it hard to see the green dots at the edges of the cyan square and how they connect to the hash table.

[W4] Missing qualitative results. Including visualisations of species’ ranges is common in this area of research (e.g. see Phillips et al, Botella et al, Dubos et al, and Cole et al) and would help this reviewer understand the impact of the “explicit” component on the generated ranges.

[W5] Some implementation details are missing in the text. For example, the value of L, the size of the hash table, and the total number of parameters. Additionally we are told that M = 8 if the "implicitness" is less than 0.5 and M = 16 if the “implicitness” is greater than 0.5. However it is unclear what value of M is used when the implicitness is exactly 0.5 which is a frequently used setting in the paper.

Additional minor comments, that do not require a response in the rebuttal:

* L40 Some issues with terminology. “Distribution range” should be either “distribution” or “range” but not both.
* L50 Add citations for iNaturalist, eBird, and PlantNet.
* L75 While the implicit neural approaches are relatively new, previous SDM approaches have used “presence only” data such as “Max Ent” (Phillips et al’s “Maximum entropy modelling of species geographic distributions” - Ecological Modelling 2006).
* L123 The task is difficult due to difficulties with the data, but what are those difficulties with the data?
* L194 I could not find where Sitzmann et al’s “Implicit Neural Representations with Periodic Activation Functions” actually states that “implicit representations perform better when the inputs are of high frequency”. Though it is possible I have missed this.
* L195 mentions wrapped encoding is for high resolution. It is actually to prevent boundary artefacts as longitude wraps from 180 to 0 (similar for latitude).
* L233 “to” should be “the”
* L309 “run our models” -> “train our models”
* L315 This paragraph is awkwardly phrased. iNaturalist is not a dataset by itself but a website.
* L621 Kim et al’s “Hybrid Neural Representations for Spherical Data” has been published in ICML 2024 and this citation should be used rather than arXiv.
* L624 Kingma and Ba’s “Adam: A method for stochastic optimization” has been published at ICLR 2015.
* L626 Rußwurm et al’s “Geographic Location Encoding with Spherical Harmonics and Sinusoidal Representation Networks” has been published in ICLR 2024.
* Figures 7, 8, 9 and 10 consist of multiple plots horizontally aligned, however the y axis is not consistent across these plots which makes it difficult to compare results. The axes labels are also very small making this issue hard to spot at first glance.

**Questions:**

[Q1] Can the authors provide more detail on the results in figure 8? What is it about the “explicit” approach that makes it so much worse in high recall situations?

[Q2] What is the justification for changing the value of M as “implicitness” changes?

[Q3] The authors note that ecological boundaries such as rivers cause sharp distinctions in ecosystems that are very close to each other geographically, and that current methods fail to model these high frequency details (see L211). Does the approach in this paper succeed at modelling this? Do range maps for appropriate species show sharp boundaries at rivers and other natural boundaries?

---

### Official Review · Reviewer_pKcJ · 2024-10-29

**Soundness:** 2
**Presentation:** 3
**Contribution:** 2
**Rating:** 3
**Confidence:** 4

**Summary:**

This paper presents a hybrid representation model for Species Distribution Modeling (SDM). It combines previous implicit model architecture FCNet (Mac Aodha+ 2019) and explicit multiresolution hashgrids (Muller+ 2022) into a novel single representation.  The motivating challenge to combine them is the challenge for implicit functions to represent local environmental information with high-frequency patterns.

The task setup is training with the popular iNaturalist data set, and testing with human labeled  S&T and IUCN datasets. However, to make it challenging for implicit functions to represent high-frequency patterns, no environmental information is used as model input.

Experiments show that the hybrid model outperforms implicit functions such as SINR, GP, BDS (table 1) and the result is not sensitive to hyper-params.

**Strengths:**

I do believe that a hybrid approach is a good direction to explore and the proposed model is reasonable

**Weaknesses:**

I still find the current study has a few issues in its setup. First, it seems unlikely that scientists want to handicap SDM models by removing the environmental information. So the main result should include environmental information, and the current result seems more like an ablation study. Second, the SOTA implicit functions have internal multi-scale representation layers (sphere2vec, Siren). They are designed to represent high-frequency patterns and should definitely be experimented in addition to FCNet.
https://www.sciencedirect.com/science/article/abs/pii/S0924271623001818
https://arxiv.org/abs/2310.06743

Furthermore, there might be room to improve on the modeling side as having an observation cap seems a waste of training examples. There are notations not defined or not mathematical in Section 3.

**Questions:**

See Weaknesses

---

### Official Review · Reviewer_MTMa · 2024-11-07

**Soundness:** 3
**Presentation:** 3
**Contribution:** 3
**Rating:** 6
**Confidence:** 2

**Summary:**

Within the field of global species density modelling, the authors propose a method for merging learnable global features with learnable local features that apply only to specific regions, enabling high-frequency features such as rivers to be provided without creating artifacts due to the continuity assumptions of neural network architectures.

**Strengths:**

The methodology is well presented, and the paper well written. I think it is an interesting approach, but it is outside my area of expertise to strongly assert the strength of the contribution to the subfield.

**Weaknesses:**

It would be good to demonstrate the utility of the method when using a variety of different cos/sin frequency representations of the global co-ordinate. When more frequencies are provided and so the model has more high spatial frequency inputs for the global representations it can use, does the boost in performance of the hybrid approach diminish?

**Minor**
- Figures are subpar. They are not high enough resolution. The text within figures should be at least 75% the size of the text in the main paper (at least $\footnotesize$). For simple figures such as line graphs which have few objects, vector graphics should be used so readers can zoom in arbitrarily far to see more detail in the plot without struggling to make things out due to rasterization blur. Yellow lines are hard to make out on the grey background and should be changed to a colour with more contrast.
- Digits in tables should be aligned by their significance (the tens places above each other, ones place above each other, etc.) to improve readability (c.f. Tables 3 and 8)
- L254 (Eq 3) min/max shouldn't be italicized (regular math has $max$ read as the product of three variables, $m$, $a$, and $x$) and should be in roman font with `$\max$` or `$\mathrm{max}$`; exp brackets should be taller with `\left( ... \right)` to go around the height of their contents.
- L160 Similarly for $(lat, lon)$, these multicharacter variables should be in roman with `\mathrm` since they are not the product of $l$, $a$, and $t$; etc.
- L309 Consider using scientific notation for numbers <0.001 to improve their readability
- L321 Consider using thousands separators for numbers >=10000
- L366 "Subsection 5.3" should be a hyperlink
- L521 "well- suited" -> "well-suited"
- Be careful with casing of titles of some references (sometimes the bib files provided by conferences don't capture acronyms correctly), e.g. "Tensorf", "roc", "Plenoctrees"
- Appendix needs to be fleshed out with text to describe tables. Table references should be hyperlinks.

**Questions:**

The authors consider the scenario where there is no additional environmental information. Would the hybrid method proposed also offer advantages when additional environmental information is available?

How exactly are global co-ordinates encoded into the network? How many cos and sin frequency representations are passed to the model?

---

### Note · Authors · 2024-11-13

I have read and agree with the venue's withdrawal policy on behalf of myself and my co-authors.